# Trunk Velocity Changes in Response to Physical Perturbations Are Potential Indicators of Gait Stability

**DOI:** 10.3390/s23052833

**Published:** 2023-03-05

**Authors:** Farahnaz Fallahtafti, Sjoerd Bruijn, Arash Mohammadzadeh Gonabadi, Mohammad Sangtarashan, Julie Blaskewicz Boron, Carolin Curtze, Ka-Chun Siu, Sara A. Myers, Jennifer Yentes

**Affiliations:** 1Department of Biomechanics, University of Nebraska at Omaha, Omaha, NE 68182, USA; 2VA Nebraska-Western Iowa Health Care System, Department of Veterans’ Affairs, Omaha, NE 68105, USA; 3Department of Human Movement Sciences, Faculty of Behavioral and Movement Sciences, Vrije Universiteit Amsterdam, 1081 HV Amsterdam, The Netherlands; 4Institute for Rehabilitation Science and Engineering, Madonna Rehabilitation Hospitals, Lincoln, NE 68506, USA; 5Department of Industrial Engineering, Amirkabir University of Technology, Tehran 15875, Iran; 6Department of Gerontology, University of Nebraska at Omaha, Omaha, NE 68182, USA; 7Department of Health & Rehabilitation Sciences, Physical Therapy Program, University of Nebraska Medical Center, Omaha, NE 68198, USA; 8Department of Health & Kinesiology, Texas A&M University, College Station, TX 77843, USA

**Keywords:** resistance, recovery, gait, stability, trunk velocity

## Abstract

Response to challenging situations is important to avoid falls, especially after medial perturbations, which require active control. There is a lack of evidence on the relationship between the trunk’s motion in response to perturbations and gait stability. Eighteen healthy adults walked on a treadmill at three speeds while receiving perturbations of three magnitudes. Medial perturbations were applied by translating the walking platform to the right at left heel contact. Trunk velocity changes in response to the perturbation were calculated and divided into the initial and the recovery phases. Gait stability after a perturbation was assessed using the margin of stability (MOS) at the first heel contact, MOS mean, and standard deviation for the first five strides after the perturbation onset. Faster speed and smaller perturbations led to a lower deviation of trunk velocity from the steady state, which can be interpreted as an improvement in response to the perturbation. Recovery was quicker after small perturbations. The MOS mean was associated with the trunk’s motion in response to perturbations during the initial phase. Increasing walking speed may increase resistance to perturbations, while increasing the magnitude of perturbation leads to greater trunk motions. MOS is a useful marker of resistance to perturbations.

## 1. Introduction

Falls are the leading cause of disability and injury in older adults [1], and most falls happen after exposure to unexpected perturbations during walking [2]. Using unexpected perturbations while walking in a laboratory allows for a controlled study of gait stability. Ultimately, a lack of gait stability will result in a fall, but it is obvious that a more useful measure of stability would not be binary. Examining responses to mediolateral perturbations is important as stability in this direction requires active control, while the balance in the anteroposterior direction is partly controlled passively by mechanics [3]. Mediolateral perturbations are more challenging to recover from than perturbations in the anteroposterior direction [4]. External moments around the center of mass can affect mediolateral stability, and the induced instability after such a perturbation can be overcome by appropriate responses [5]. Measuring stability of the person walking can be captured by responses to physical perturbations [6,7,8]. 

Responses to perturbations can be influenced by the direction, timing, and magnitude of the perturbation, in addition to walking speed [9]. Different strategies (e.g., ankle strategy and stepping strategy [9]) can be used to maintain stability during walking. These strategies lead to changes in the body’s momentum, executed through muscle activations and limb motion [10]. Lateral platform perturbations are easier to compensate for than medial perturbations, as they allow for better exploiting the base of support [5]. Medial platform perturbations induce a combination of crossover and stepping strategies. The recovery strategy after a perturbation is affected by the duration of- and timing within the gait cycle [11]. In steady-state gait, walking speed affects gait parameters, which may play a role in gait stability [12]. Cadence and step length decrease significantly with decreased walking speed [13]. At faster speeds, decreased interlimb coordination, margin of stability (i.e., a measure of mechanical stability), and variability have been reported [14,15,16].

To avoid falls, the regulation of trunk velocity is essential because of its mass, notable height above the feet (base of support), and its impact on the center of mass movement. The trunk contains almost 50% of the total body weight or two-thirds when including the head and arms [17]. Trunk movement following a perturbation has demonstrated different aspects of recovery capacity than the margin of stability (MOS) [18]. However, in the abovementioned study, the anteroposterior distance between the center of mass and the leading limb toe was defined as the margin of stability. In the current study, MOS was defined to describe gait stability in the mediolateral (ML) direction [19,20]. Mediolateral stability is achieved by placing the foot at a certain distance laterally to the (extrapolated) COM, thereby redirecting its movement [19]. MOS variability is defined as step-to-step variations in the magnitude of MOS.

How the responses to perturbations are associated with a gait stability metric under challenging conditions has not been the topic of many studies. We studied the effects of walking speed and perturbation magnitude on responses following a perturbation and whether these responses correlated to the margin of stability. We used the method of Bruijn et al. [7] to divide the responses to the perturbations into two phases. We hypothesized that the initial phase of the response would be associated with the average magnitude of MOS due to the quickest attempt to adjust the base of support. We hypothesized that the recovery phase would be associated with changes in MOS variability in the subsequent steps after perturbations. We also hypothesized the initial phase response to be smaller at higher walking speeds due to increasing body momentum. We expected to observe a detrimental effect of increasing perturbation magnitude on MOS during both phases of the responses to the perturbation.

## 2. Materials and Methods

### 2.1. Subjects

Eighteen healthy, young participants aged 19–35 years were recruited. All participants were physically active, within normal weight ranges (body mass index of 18.5 to 24.9 kg/m^2^) and without neurological or musculoskeletal issues. All subjects signed the consent form and were screened before testing. The University’s Institutional Review Board reviewed and approved all procedures. The study was conducted according to the guidelines of the Declaration of Helsinki and approved by the Institutional Review Board (or Ethics Committee) of the University of Nebraska Medical Center (protocol code 139-19-EP).

### 2.2. Setup

Data were collected using a CAREN-Extended system (Computer-Assisted Rehabilitation Environment, Motek Medical, Amsterdam, The Netherlands). The CAREN-Extended system configuration includes an integrated 10-camera, motion capture system (MX T20S, Vicon Motion Systems, Inc.; Oxford, UK), a six-degree-of-freedom motion platform (Sarnicola Simulation Systems Inc.; Conklin, New York, NY, USA), and an instrumented treadmill (Bertec Corp; Columbus, OH, USA). D-Flow software (version 3.20.0, Motekforce link, Houten, The Netherlands) was used to control hardware components. A harness was worn for confidence and safety. Kinematic data were captured at 100 Hz.

### 2.3. Procedures

Subjects were asked to wear a form-fitting suit, and the human body model (HBM-Gait) marker set with 28 passive retroreflective markers for the trunk, pelvis, leg, and feet were used to record the motion of body segments [21]. Markers were placed on the right and left acromion, seventh cervical vertebra, 10th thoracic vertebra, the xiphoid process of the sternum, the jugular notch of the sternum, right and left anterior superior iliac spines, right and left posterior superior iliac spines, right and left lateral and medial epicondyle of the knees, right and left thighs and shanks, right and left medial and lateral malleolus of the ankles, right and left heels, and right and left second and fifth metatarsals. 

Subjects were asked to find their preferred walking speed on a treadmill (Figure 1). The preferred speed was estimated as the treadmill operator increased or decreased speed until the preferred speed was found [12]. To determine the preferred walking speed, participants started walking at 0.67 m/s (1.5 mph). Then, we asked them to let us know if we needed to increase or decrease the speed based on their level of comfort. Once they indicated that they had reached a speed they were comfortable with, the speed was increased by 0.22 m/s (0.5 mph). They were asked to walk at this new speed for 1 min, and we asked about their comfort level. If they indicated that the speed was too fast, the speed was slightly decreased (0.1 m/s). If they indicated that the faster speed was still comfortable, we increased the speed again (0.1 m/s). This process was repeated until they reached a comfortable walking speed. We asked them to walk uninterrupted for 1 min to ensure this was the correct speed. We chose their preferred speed. This process could take up to 10 min. After the preferred speed was determined, the participants rested for 5 min. After the preferred walking speed was determined, participants were asked to walk at three different speeds (slow, preferred, and fast). On the basis of prior pilot work in our laboratory, fast speed was defined as +40% and slow speed was defined as −40% of the participant’s preferred walking speed. Three 45 s walking trials at each speed were performed as a baseline without perturbation. 

After baseline trials were completed, perturbation trials started. To create a perturbation, one platform translation was randomly applied at either right or left heel contact (Table 1). 

To eliminate the effect of learning and to prevent anticipation of only one translation direction, translations to the right and left sides were randomized. Platform translations to the right side were repeated three times, and platform translations to the left side were performed one time. Platform translations that were applied to the right side at right heel contact were defined as lateral perturbations and at left heel contact were considered medial perturbations. Only perturbations to the right side at left heel contact were used for data analysis. To remove the effect of first exposure to the perturbation for each participant, we removed the first perturbed trial for every subject walking at a preferred speed with a large lateral perturbation. Platform translations were triggered at heel contact, detected in real time using the marker-based algorithm under the human body module developed by Motek. The platform translations were delivered randomly between the 25th and 35th seconds of each trial (Figure 2). The translation was designed to be applied in ~0.7 s for all conditions. The maximum acceleration and velocity of the platform translations were 1.24 m/s^2^ and 0.17 m/s, respectively. A minimum of 1 min rest between every five trials was given.

To evaluate the effect of walking speed, participants were asked to walk at the three speeds (slow, preferred, and fast defined above) with a medium platform translation (5 cm) during all speed conditions. Three platform translation magnitudes were used during the preferred walking speed to determine the effect of perturbation magnitude. Three ratios (1/3, 2/3, and 1) of the maximum displacement of the platform, which was 7.5 cm, were used to create small, medium, and large perturbations. 

### 2.4. Data Processing and Analysis

Kinematic (marker) data were tracked in Nexus software (version 2.7, Vicon Motion Systems Inc., Oxford, UK) and exported as a Visual 3D file (Visual 3D, C-Motion, Germantown, MD, USA) for further processing in MATLAB 2019b software (Mathworks Inc., Natick, MA, USA). The first 5 s of each trial was removed to remove the effect of non-stationarity. 

The local minimum position in the vertical trajectory of heel markers from the remaining 40 s of data was used to identify the heel contacts. Detected heel contacts were visually inspected and corrected when needed [7]. 

#### 2.4.1. Trunk’s Response

Responses to the perturbation were calculated as proposed by Bruijn et al. [7]. To this aim, linear and angular velocities of the trunk were used to create a limit cycle signal for each subject and trial. Therefore, velocity in six dimensions was calculated (d) (Figure 3). 

The unperturbed velocity signals were derived from the baseline conditions, performed at three walking speeds. The perturbed signal (to the right side) was derived at the moment that the left heel contacts were detected to the fifth heel contact after the perturbation (ipsilateral to the last heel contact before perturbation) (Figure 4). 

According to pilot data analysis, five strides were sufficient to return to steady state walking for all conditions. For each percentage in this limit cycle, we calculated the normal variability for each dimension using standard deviation ($v_{NW}$). The Euclidean distance between the perturbed velocity signal (PW) and the unperturbed velocity signal (NW) was computed as
(1)$$D{\left(k\times 100+i\right)}_{i=1:100}^{k=0:n-1}=\sqrt{\overset{6}{\underset{d=1}{\sum }}{\left(\frac{\mathrm{NW}{\left(i\right)}_{d}-\mathrm{PW}{\left(k\times 100+i\right)}_{d}}{v_{\mathrm{NW}}{\left(i\right)}_{d}}\right)}^{2}},$$
where D(k_100 + i) is the normalized distance (unit: number of standard deviations) for i% of stride k + 1 (where “n” represents the maximum number of strides during PW), dimension number is shown by d, NW is the unperturbed walking trial, PW is the state of the perturbed walking trial, and $v_{\mathrm{NW}}$ is the variance of the unperturbed walking trial (limit cycle). We detected the time to the maximum D after perturbation (τ, Figure 5). The exponential decay toward an unperturbed cycle was quantified using the following equation: (2)$$D\left(i\right)=A+\left(B-A\right)\times e^{\left(-\beta \left(i-\tau \right)\right)},$$
where D indicates the distance of the perturbed gait cycle from the average unperturbed limit cycle, A indicates the relaxation distance (steady state), B refers to the maximum distance after perturbation, τ refers to the time from the onset of perturbation to the maximum distance, and β refers to the rate of return to the unperturbed cycle.

The distance between perturbed and unperturbed trunk velocity was then divided into two phases: (1) the initial phase and (2) the recovery phase (Figure 5). The initial phase of response was quantified using the maximum distance after perturbation (B) and the time from the onset of perturbation to the maximum distance (τ). The recovery phase was quantified in two ways. First, an exponential function was fitted to the data, from which β was derived as the exponential decay quantifying the rate of return towards the unperturbed cycle [22]. Second, the distance from the unperturbed gait cycle at the first ipsilateral heel contact (D_hc_) during the recovery phase was quantified (Figure 5). A shorter time from the onset of perturbation to the maximum distance from the unperturbed gait pattern (shorter τ) implies a greater acceleration, and a smaller maximum distance after perturbation (shorter B) indicates more resistance in response to the perturbations. The preferred performance during the recovery phase would be a greater rate of return toward the unperturbed cycle (steeper β) and a smaller distance at the first recovery heel contact (shorter D_hc_).

#### 2.4.2. Stability Metric

The MOS was determined by the distance between the border of the base of support (in our study, the heel marker position at heel contact) and the extrapolated center of mass [20]. The extrapolated center of mass was defined by the center of mass position plus its velocity divided by √𝑔/𝑙 in which 𝑔 is the acceleration of gravity and 𝑙 is the effective pendulum length defined as the distance from the center of mass to the lateral right and left ankle markers at heel contact. 

MOS in the ML direction were calculated on the immediate next heel contact after perturbation (MOS_1_). The mean and standard deviation of the MOS at heel contact for five complete strides after perturbation were calculated (MOS mean, MOS variability) during each walking trial.

### 2.5. Statistics

Assumptions of normality were examined for each dependent variable using the Shapiro–Wilk test. To test our first hypotheses that increasing walking speed would affect the trunk responses during the initial and the recovery phases, we used a one-way repeated-measure ANOVA (1 × 3) to compare each variable (B, τ, β, and D_hc_) across three walking speeds for medial perturbations. To test our second hypothesis that increasing perturbation magnitudes would deteriorate the responses at initial and recovery phases, we used a one-way repeated-measure ANOVA (1 × 3) to compare each variable (B, τ*,* β, and D_hc_) across three magnitude conditions for medial perturbations. To assess the association between response variables (B, τ, β, D_hc_) and measures of stability, separate linear mixed models were run, one for each stability measure (MOS_1_, MOS mean, and MOS variability), with each trunk response as an individual outcome (B, τ, β, and D_hc_). In the models, we separately included speed or magnitude as independent factors. To examine if the association between MOS measures and the trunk responses differed on the basis of the levels of speed/magnitudes, in each model, the interactions between dependent variables and speed/magnitude factors were analyzed. All analyses were performed using SPSS version 23 (IBM Corp., Armonk, New York, NY, USA). Statistical significance was set at α = 0.05.

## 3. Results

Eighteen healthy young adults completed this study (Table 2). The results of the repeated-measure ANOVA and regression analysis models are provided below.

The maximum distance for the attractor (B) increased when the walking speed decreased (*p* < 0.0001). B was significantly greater during slow compared to preferred and fast walking speed conditions (*p*’s < 0.0001), and B was also greater during preferred compared to fast walking speed conditions (*p* = 0.008) (Figure 6).

The maximum distance for the attractor (B) and the distance at the first heel contact (D_hc_) significantly increased when the magnitude of perturbation increased (*p* < 0.0001). B and D_hc_ were greater during large perturbations when compared to medium and small perturbations, and the medium was also significantly different than small (*p* < 0.0001). The rate of return to the attractor, β increased with decreasing perturbation magnitude (*p* = 0.029). β was quicker during small compared to large perturbations (*p* = 0.005) (Figure 7).

### 3.1. Association of Trunk Responses with MOS Measures (Effect of Speed)

We found a positive association between MOS variability and τ (*p* = 0.041, estimate value = 0.031). MOS mean was positively associated with B (*p* = 0.001, estimate value = 0.036). MOS_1_ and MOS mean were associated with D_hc_ (negatively *p* = 0.018, estimate value = −0.026, and positively *p* = 0.004, estimate value = 0.041, respectively). None of the MOS measures were associated with β. Complete results can be found in Table 3 and Table 4.

### 3.2. Association of Trunk Responses with MOS Measures (Effects of Magnitude)

We found a significant association between MOS variability and τ (*p* = 0.02, estimate value = 0.031). MOS mean was associated with B, independent of magnitude (*p* = 0.004, estimate value = 0.306). Furthermore, none of the MOS measures were associated with β and D_hc_. Complete results can be found in Table 5 and Table 6.

## 4. Discussion

The purpose of this study was to evaluate the effects of walking speed and perturbation magnitude on trunk velocity changes from the steady state in response to medial perturbations. Moreover, we sought to determine the association between responses to perturbations and measures of dynamic stability (i.e., MOS). The perturbations were generated using translations of a platform to the right direction at left heel contacts (medial perturbations). We expected to observe significant effects of speed and magnitude on the responses to the perturbations. It was hypothesized that the initial phase of trunk response would be associated with the average values of MOS, while the recovery phase corresponded more to the MOS variability. This hypothesis was partially supported, as we indeed found an association between MOS mean and maximum deviation from the steady state. Moreover, we hypothesized that, with increasing walking speed, we would observe changes in the initial and recovery phases of the trunk’s motion in response to perturbation. Although walking speed affected the initial phase of the response, the recovery was not significantly affected. A detrimental effect of increasing the perturbation magnitude on both phases of perturbation responses was detected for medial perturbations. 

MOS mean was associated with the maximum distance from steady state after medial perturbations across speed and magnitude conditions. Medial perturbations caused the base of support to move toward the extrapolated center of mass [23]. It has been shown that after a perturbation, stability is recovered by taking faster, shorter, and wider steps [23]. However, the stepping responses might be different based on the perturbation type. Our results showed that the maximum deviation of trunk velocity from steady state in response to a perturbation was not necessarily predicted by MOS at the first heel contact. The association between average MOS in the subsequent steps was a better indicator of the maximum changes in trunk deviation. Moreover, smaller changes in the base of support compared to changes with extrapolated center of mass may have caused decreased MOS after perturbation. This behavior explains the negative association between MOS at first heel contact and the trunk velocity distance from attractor at first heel contact. 

Our hypothesis that MOS variability would be associated with the recovery phase of the perturbation response was rejected; however, we observed an association between MOS variability with the initial phase of trunk response (i.e., time to the trunk maximum distance from the steady state). MOS variability was calculated on the basis of the variability of foot placement with respect to the extrapolated center of mass. Prior work has shown that, after mediolateral perturbations, foot placement adjustments are strongly affected by the perturbation timing, magnitude, and direction [24]. The closer the perturbation is to the heel contact, the more the strategies vary between subjects [24]. Therefore, it is likely that our subjects used multiple strategies, which led to MOS variability not having a consistent relationship with trunk velocity deviation during the recovery phase. The results of this study indicated that the faster it gets to the maximum distance from the attractor, the more variable the MOS pattern was. Considering this relationship, the association between the initial response time and fall risk could be an interesting topic for future studies. 

### 4.1. Effects of Speed

In accordance with our hypothesis, the initial response to mediolateral perturbations improved with increasing walking speed, considering a smaller trunk velocity distance from the steady state. During faster speeds, greater momentum of the body may have led to greater resistance to deviations from steady state. Increasing walking speed seemed to increase body momentum, preventing walking from becoming unstable as the body progressed forward quickly after initial contact [25]. Timing (τ) was not affected by walking speed; this could be related to the perturbation type in this study. We designed the perturbation to be applied in approximately the same duration for all speeds. Therefore, the velocity and acceleration of platform translation were similar between speeds. Equal acceleration and translation could have been a reason for unchanged τ across speeds. Bruijn et al. [7] found a significant effect of walking speed on τ after perturbations; however, in that study, the perturbations were force-controlled perturbations, whereas we used position-controlled perturbations.

The recovery phase was not affected by speed. During outward waist pulling, trunk rotation showed a substantially greater deviation from the steady state compared with the inward perturbation (comparable with medial in this study) [26], which could be a reason for not observing differences after medial perturbations. According to Roeles et al. [23], mediolateral translation of the base of support is expected to have a similar effect on stability as a mediolateral waist perturbation in the opposite direction [23]. In the study by Hof et al. [26], after pushes to the right or left at waist level, the foot was placed away from the push. The recovery strategy is based on swing leg control. Adduction of the right leg, possibly by the adductor magnus or gracilis, has been reported as a crossover step strategy in response to an outward push [27] (comparable to medial platform perturbation in this study). However, these strategies were not always seen in this study, which might be related to the magnitude of perturbation and the flexibility of healthy individuals to respond in various ways to a perturbation.

The largest perturbation intensity used in this study was a translation of 7.5 cm, applied in a constant amount of time. At this perturbation intensity, noticeable trunk movement was observed compared to the lowest magnitude (2.5 cm). Healthy participants in this study coped with the smallest magnitude of perturbation with the least deviation from the steady state. This may be why changes in the responses were revealed across different perturbation levels. However, across different walking speeds, a medium magnitude of perturbation was applied. With changes in walking speed, different balancing strategies such as trunk rotation, counter rotation of upper limbs, and leg movements could be employed [28,29], which may have resulted in inconsistent and variable between-subject responses. 

### 4.2. Effects of Magnitude

Responses of the trunk in the initial phase were affected by perturbation magnitude. Smaller perturbations led to a smaller maximal deviation from steady state, in line with our hypothesis. Following a medial foot placement, a hip adductor moment is needed to swing the leg more medially [27]. By increasing the magnitude of the perturbation, possible hip adductor moment elevation (leading to flexor/extensor moment) [30] might have led to greater trunk velocity and subsequently increased the max deviation distance from steady state. Foot placement was not the only strategy taking part in stabilization during our experiment as we did not limit the upper body movement in our participants, which may have also played a role in responses. However, the time to the maximal distance from steady state showed no significant differences between perturbation magnitudes. Although an increased magnitude of perturbation most likely led to increased momentum in the mediolateral direction, increasing the acceleration of platform translation may nullify the timing of the initial response. Moreover, the time it takes to the max distance during the initial phase of response is related to body mass index and resistance to the perturbation [7], which remains constant across different magnitudes of perturbation.

During small perturbations, the rate of recovery was faster, meaning that subjects were quicker to return to a steady state during small magnitude conditions. Muscle activity contributing to balance recovery was shown to be scaled with the magnitude and direction of perturbation [30]. By increasing the magnitude of perturbation, more muscle activity is required, which contributes to a longer time to activate and recover from a perturbation. Increasing the magnitude of a perturbation potentially increases the body momentum in the mediolateral direction and, consequently, leads to greater maximum deviation from the steady state; therefore, it takes longer to return to the steady state as the walking speed and forward momentum remain constant.

On the recovery side, the distance from the unperturbed gait cycle at the first ipsilateral heel contact showed a larger distance from steady state at larger perturbation magnitudes. This is in line with previous research that demonstrated stepping action increased with the perturbation magnitude [26]. The larger the perturbation magnitude, the faster the swing leg accelerated to place the foot toward the fall direction, which may have led to increased trunk velocity at first heel contact after perturbation for medial perturbations.

This study had some limitations. We investigated perturbations occurring at one specific time during the gait cycle and in a particular direction, leaving it unknown if our results are generalizable to perturbations occurring at other phases of the gait cycle and/or in other directions. The duration of perturbations was equal across conditions; with walking speed changes, the perturbation ended in different phases of the gait cycle. 

## 5. Conclusions

In conclusion, the MOS mean measure was associated with the initial responses. We suggested that the trunk’s response may lead to a combination of coping strategies [5,31]. Walking speed and perturbation magnitude affect the trunk’s responses (magnitude more so than speed). Walking faster is more resistant to perturbations. Increasing the magnitude of perturbation can negatively affect the initial and recovery responses to medial perturbations. This should be kept in mind as an essential fact that could explain the strategies being used to avoid falls under different perturbed circumstances. 

## Figures and Tables

**Figure 1 sensors-23-02833-f001:**
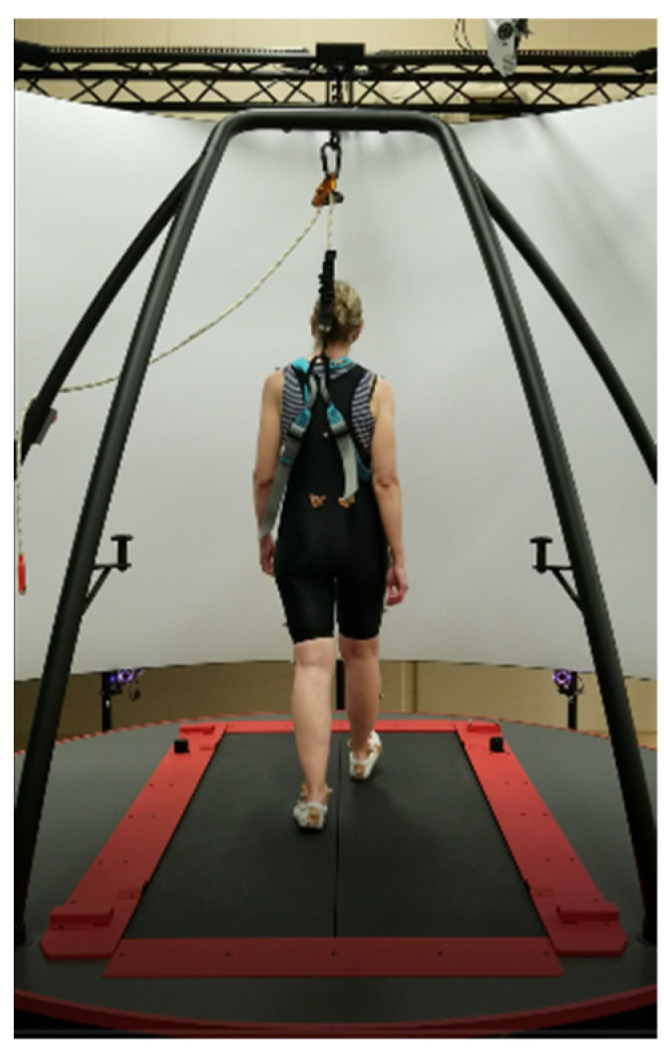
An example of the experimental setup with one individual using the CAREN system. The platform can be translated in the mediolateral direction during a participant walking on the treadmill in the anteroposterior direction.

**Figure 2 sensors-23-02833-f002:**
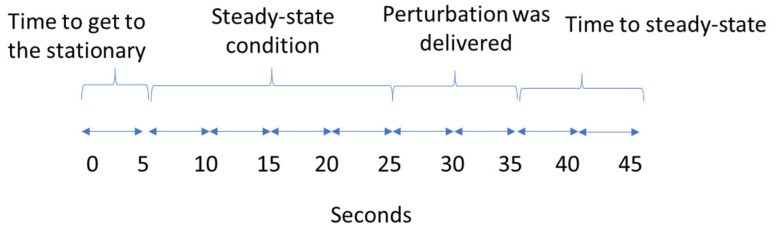
Duration of one trial (45 s) and division to each considered period.

**Figure 3 sensors-23-02833-f003:**
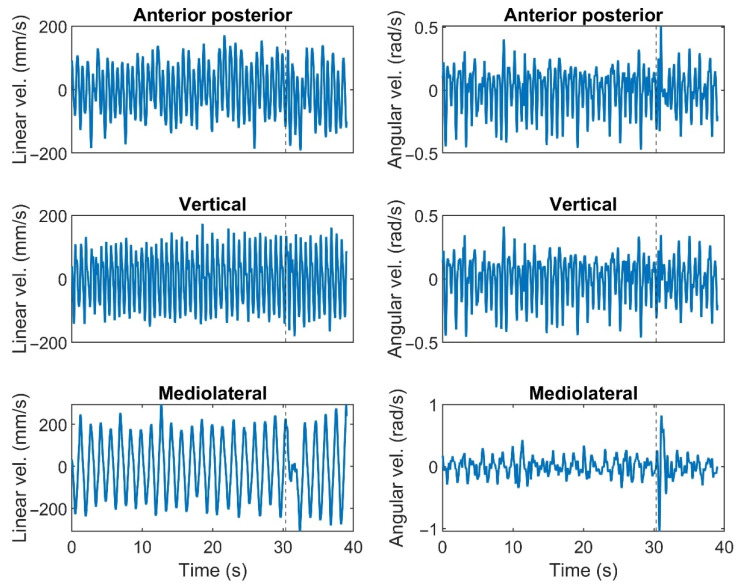
Exemplar linear (**left column**) and angular (**right column**) trunk velocity trajectories in response to the medium perturbation during walking at slow speed. The dashed line indicates the perturbation time. Translation directions and axis of rotations were indicated on top of the linear and angular velocities, respectively.

**Figure 4 sensors-23-02833-f004:**
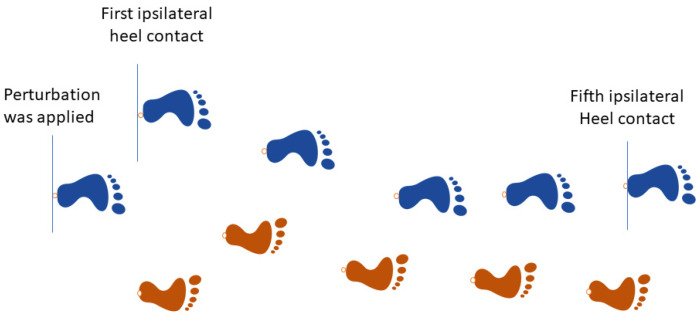
The representative steps are used for the calculation of the perturbed velocity signal. The steps were perturbed by the platform translation to the right side at the left heel contact.

**Figure 5 sensors-23-02833-f005:**
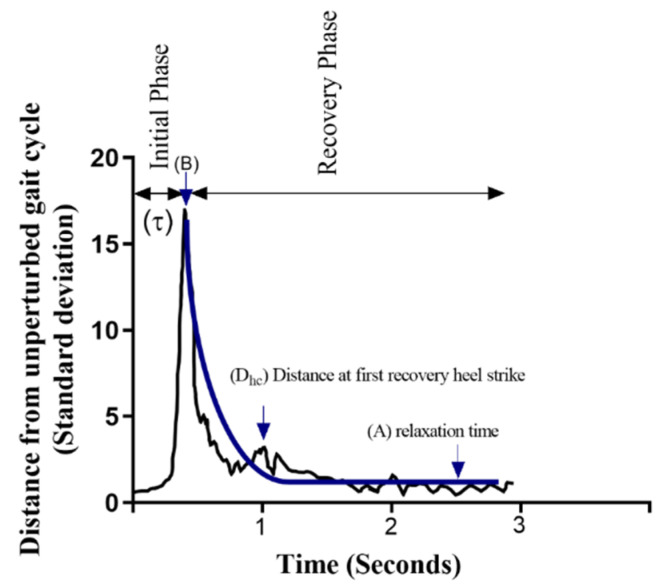
Explanation of compensatory responses. The *x*-axis denotes the time from the onset of perturbation. The *y*-axis is the standard deviation of the distance between the trunk velocity vector during perturbed and unperturbed gait cycles. A is relaxation time, B is the maximum deviation from the unperturbed gait cycle after perturbation, τ is the time to maximum distance, β is the rate of return to the unperturbed gait cycle, which is derived by fitting an exponential curve into the distance from the perturbed gait cycle trajectory starting at B toward the unperturbed gait cycle, and D_hc_ is the distance from the unperturbed gait cycle of the first recovery heel contact after perturbation. Figure adapted from [7].

**Figure 6 sensors-23-02833-f006:**
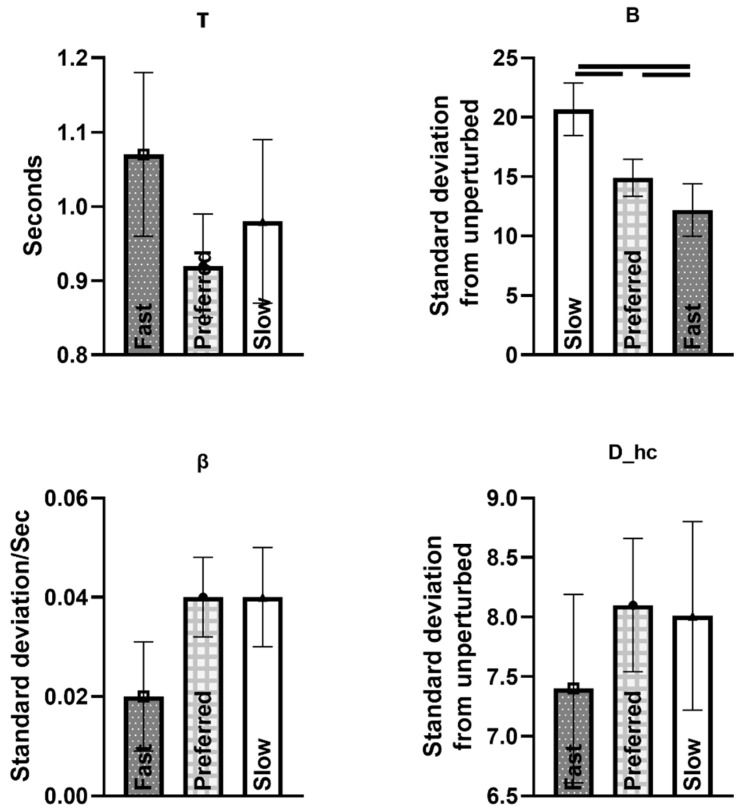
Trunk velocity changes from steady state (number of standard deviations away from steady state, which is the distance between the trunk velocity vector during perturbed and unperturbed gait cycles) in response to medial perturbations at medium perturbation magnitude. Comparisons were made between walking speeds.

**Figure 7 sensors-23-02833-f007:**
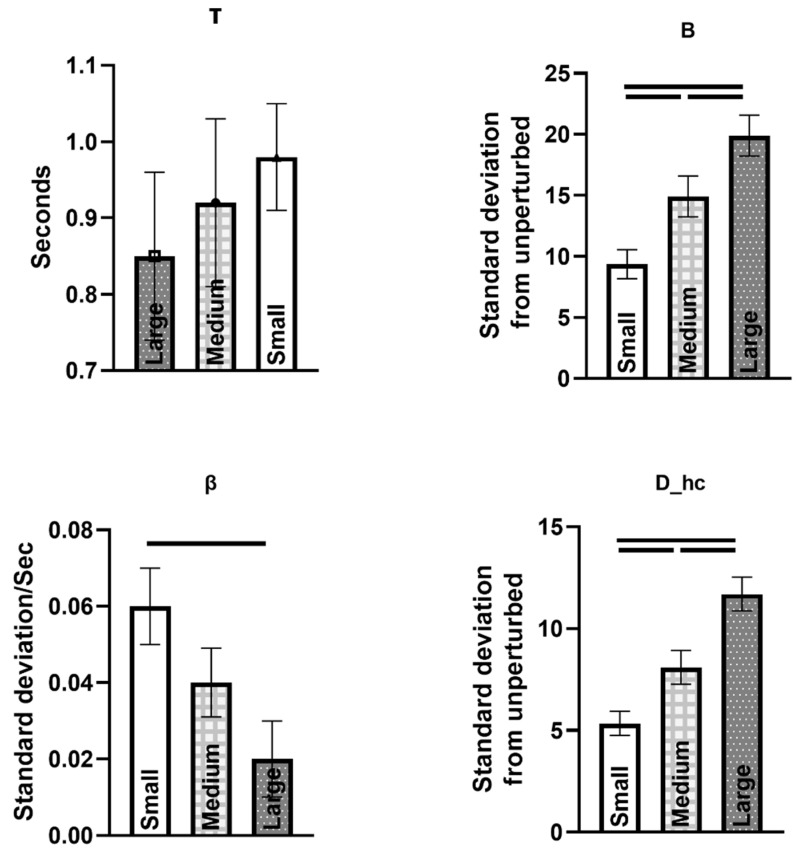
Trunk velocity changes from steady state (number of standard deviations away from steady state, which is the distance between the trunk velocity vector during perturbed and unperturbed gait cycles) in response to medial perturbations at preferred walking speed. Comparisons were made between magnitudes of perturbation.

**Table 1 sensors-23-02833-t001:** A combination of speeds and magnitudes of platform translations was applied at right and left heel contacts in a randomized order. There were a total of five walking conditions containing four trials, three trials to the right side and one trial to the left side. Only right-side translations were used for data analysis. Platform translations to the right at right heel contact were defined as lateral perturbations and at left heel contact were considered medial perturbations. Only medial perturbations were used for processing.

	Small (2.5 cm)	Medium (5 cm)	Large (7.5 cm)
Slow Speed (−40% preferred)		Condition #2	
Preferred Speed	Condition #1	Condition #3	Condition #5
Fast Speed (+40% preferred)		Condition #4	

**Table 2 sensors-23-02833-t002:** Demographic data for the participants. All values are written as mean (standard deviation).

Age (Years)	Body Mass (kg)	Height (cm)	Preferred Walking Speed (m/s)
24.26 (3.67)	67.71 (10.57)	174.48 (10.37)	1.22 (0.06)

**Table 3 sensors-23-02833-t003:** Significant associations at the fixed magnitude of perturbation across walking speed conditions were observed between parameters of trunk responses and stability metrics. B is the maximum deviation from the unperturbed gait cycle after perturbation. τ is the time to maximum distance. β is the rate of return to the unperturbed gait cycle, which is derived by fitting an exponential curve into the distance from the perturbed gait cycle trajectory starting at B toward the unperturbed gait cycle. D_hc_ is the distance from unperturbed gait cycle of the first recovery heel contact after perturbation.

Covariates and Interactions	Dependent Variables							
	τ		B		D_hc		β	
	F	sig	F	sig	F	sig	F	sig
MOS1	2.010	0.164	1.222	0.275	6.043	0.018 *	0.044	0.834
MOS1 × speed	0.035	0.966	0.545	0.584	0.412	0.665	1.634	0.207
MOS mean	2.387	0.130	11.747	0.001 *	9.465	0.004 *	0.223	0.639
MOS mean × speed	0.009	0.991	0.260	0.772	1.163	0.322	0.819	0.448
MOS variability	4.425	0.041 *	1.666	0.204	0.778	0.383	0.018	0.895
MOS variability × speed	0.218	0.805	0.324	0.725	0.438	0.648	0.717	0.494

* Significant (*p*-value < 0.05).

**Table 4 sensors-23-02833-t004:** Summary of the mixed models at the fixed magnitude of perturbation across walking speeds for significant associations observed between parameters of trunk responses and stability metrics. Preferred speed is considered as a baseline. B is the maximum deviation from the unperturbed gait cycle after perturbation. τ is the time to maximum distance. β is the rate of return to the unperturbed gait cycle, which is derived by fitting an exponential curve into the distance from the perturbed gait cycle trajectory starting at B toward the unperturbed gait cycle. D_hc_ is the distance from the unperturbed gait cycle of the first recovery heel contact after perturbation.

Dependent	Factors and Covariates	Speed (Levels)	Estimated	Std. Error	T	Sig	95% Confidence Interval	
							Lower Bound	Upper Bound
τ	MOS variability	-	0.031	0.022161	-	-	−0.0135	0.075951
	Speed	Fast	0.657	0.512	1.283	0.206	−0.376	1.690
		Slow	0.521	0.504	1.035	0.306	−0.495	1.538
	MOS variability × speed	Fast	−0.015	0.025	−0.616	0.541	−0.065	0.035
		Slow	−0.016	0.027	−0.612	0.544	−0.071	0.038
B	MOS mean		0.306	0.139	2.199	-	0.025	0.587
	Speed	Fast	−6.049	9.293	−0.651	0.519	−24.804	12.705
		Slow	3.852	9.139	0.421	0.676	−14.592	22.297
	MOS mean × speed	Fast	−0.125	0.175	−0.714	0.479	−0.478	0.228
		Slow	−0.063	0.184	−0.342	0.734	−0.435	0.309
D_hc	MOS1	-	−0.026	0.047	-	-	−0.122	0.069
	MOS mean	-	0.041	0.051	-	-	−0.061	0.143
	Speed	Fast	−1.201	3.389	−0.354	0.725	−8.039	5.638
		Slow	3.594	3.333	1.079	0.287	−3.131	10.320
	MOS1 × speed	Fast	−0.050	0.057	−0.881	0.384	−0.164	0.065
		Slow	−0.027	0.055	−0.487	0.629	−0.137	0.084
	MOS mean × speed	Fast	0.090	0.064	1.414	0.165	−0.038	0.219
		Slow	0.026	0.067	0.390	0.698	−0.109	0.162
β	Speed	Fast	−0.006	0.053	−0.128	0.899	−0.115	0.101
		Slow	0.003	0.052	0.072	0.943	−0.103	0.110

* Significant (*p* < 0.05).

**Table 5 sensors-23-02833-t005:** Significant associations during preferred walking speed across different magnitudes of the perturbation conditions were observed between the parameters of the trunk’s responses and stability metrics. B is the maximum deviation from the unperturbed gait cycle after perturbation. τ is the time to maximum distance. β is the rate of return to the unperturbed gait cycle, which is derived by fitting an exponential curve into the distance from the perturbed gait cycle trajectory starting at B toward the unperturbed gait cycle. D_hc_ is the distance from unperturbed gait cycle of the first recovery heel contact after perturbation.

Covariates & Interactions	Dependent Variables							
	τ		B		D_hc		β	
	F	sig	F	sig	F	sig	F	sig
MOS1	3.451	0.070	2.395	0.129	2.329	0.134	0.729	0.398
MOS1 × magnitude	1.262	0.294	0.283	0.755	0.277	0.759	0.103	0.902
MOS mean	3.233	0.079	9.042	0.004 *	2.201	0.145	0.031	0.862
MOS mean × magnitude	0.598	0.555	0.534	0.590	0.088	0.916	0.147	0.864
MOS variability	5.803	0.020 *	1.846	0.182	0.129	0.721	0.345	0.560
MOS variability × magnitude	0.389	0.680	0.375	0.690	0.249	0.781	0.057	0.945

* Significant (*p* < 0.05).

**Table 6 sensors-23-02833-t006:** Summary of the mixed models at fixed walking speed across perturbation magnitude conditions for significant associations observed between parameters of trunk responses and stability metrics. A medium magnitude of perturbation is considered as a baseline. B is the maximum deviation from the unperturbed gait cycle after perturbation. τ is the time to maximum distance. β is the rate of return to the unperturbed gait cycle, which is derived by fitting an exponential curve into the distance from the perturbed gait cycle trajectory starting at B toward the unperturbed gait cycle. D_hc_ is the distance from the unperturbed gait cycle of the first recovery heel contact after perturbation.

Dependent	Factors and Covariates	Magnitude (Levels)	Estimated	Std. Error	T	Sig	95% Confidence Interval	
							Lower Bound	Upper Bound
τ	MOS variability	-	0.031	0.0208	-	-	−0.011	0.073
	Magnitude	Small	0.452	0.528	0.855	0.397	−0.614	1.518
		Large	0.117	0.485	0.241	0.811	−0.863	1.097
	MOS variability × magnitude	Small	0.012	0.031	0.381	0.705	−0.050	0.074
		Large	−0.015	0.030	−0.520	0.606	−0.075	0.044
B	MOS mean	-	0.306	0.114	-	-	0.075	0.536
	Magnitude	Small	−4.076	8.367	−0.487	0.629	−20.961	12.808
		Large	2.021	7.691	0.263	0.794	−13.499	17.542
	MOS mean × magnitude	Small	−0.133	0.167	−0.804	0.426	−0.470	0.202
		Large	−0.159	0.168	−0.949	0.348	−0.499	0.179
D_hc	Magnitude	Small	−0.857	4.325	−0.198	0.844	−9.586	7.871
		Large	4.852	3.976	1.220	0.229	−3.171	12.876
β	Magnitude	Small	0.057	0.068	0.833	0.409	−0.081	0.195
		Large	−0.021	0.063	−0.330	0.743	−0.148	0.106

## Data Availability

The data presented in this study are available on request from the corresponding author.

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
