# Peer review of "Trunk Velocity Changes in Response to Physical Perturbations Are Potential Indicators of Gait Stability"

_sensors, 2023, doi:10.3390/s23052833_

Round 1

Reviewer 1 Report

This study attempted to understand the effect of lateral perturbation on margin of stability. To understand the balance control in the medial-lateral direction may help to develop a rehabiliation protocol. Although I think a couples of scholars published similar manuscripts, this study has advantages to use margin of stability, the quantative tool for anwer this research question about walking on diffenret walking conditions. This manuscript is well-written and very interesting. I only have one minor equation: why authors select the -40% and 40% as slow and fast walking speed?

Author Response

We would like to thank the reviewers for their helpful comment. We have responded to the comment, point-by-point, below.

Reviewer 1:

This study attempted to understand the effect of lateral perturbation on margin of stability. To understand the balance control in the medial-lateral direction may help to develop a rehabiliation protocol. Although I think a couples of scholars published similar manuscripts, this study has advantages to use margin of stability, the quantative tool for anwer this research question about walking on diffenret walking conditions. This manuscript is well-written and very interesting. I only have one minor equation: why authors select the -40% and 40% as slow and fast walking speed?

Thank you for the great comment. In our previous studies, we tested the differences in  margin of stability across different walking speeds with ±20% and ±40% of the preferred walking speed in clinical and healthy populations, respectively (1, 2). According to the results of the previous studies and the pilot tests, we found that ±40% speed change were needed to challenge healthy individuals enough to create stability differences across walking speeds during perturbed conditions.

References

  1. Fallahtafti F, Curtze C, Samson K, Yentes JM. Chronic obstructive pulmonary disease patients increase medio-lateral stability and limit changes in antero-posterior stability to curb energy expenditure. Gait & Posture. 2020;75:142-8.
  2. Fallahtafti F, Mohammadzadeh Gonabadi A, Samson K, Yentes JM. Margin of Stability May Be Larger and Less Variable during Treadmill Walking Versus Overground. Biomechanics. 2021;1(1):118-30.

Reviewer 2 Report

In this work, authors studied the effect of mediolateral physical perturbations on the trunk response and gait stability as function of walking speed.

The experimental protocol was described with details but no figure of the experimental setup was included. Results need more clarification. The manuscript was written with good quality but some notations should be unified between the text, tables and figures. In the following, you can find few points that should be addressed:

1/ Line 112:  “Subjects were asked to find their preferred walking speed on a treadmill…”: please include the distance of the walking path in this paragraph.

2/ Line 95-102: please add a figure including photos of the experimental setup.

3/ Line 134: “Only perturbations to the right side at left heel contact were used for data analysis”: Why the analysis was limited only to this side?

4/ Line 163-169: Please add figures to show examples of kinematics signals of the trunk, before, after and during the perturbation.

5/ Line 165: “Therefore, velocity in six dimensions were calculated (d)”: what is the significance of “(d)” and is it used elsewhere in the manuscript?

6/ Line 197-184: please define each parameter in a separate line.

7/ In Figure 4, Figure 5, table 3, table 4, table 5 and table 6: please unify and use the same symbols for the parameters tau (τ) and beta (β) as are used in the text.

Author Response

We would like to thank the reviewer for the helpful comments. We have responded to each one, point-by-point below.

Reviewer 2:

In this work, authors studied the effect of mediolateral physical perturbations on the trunk response and gait stability as function of walking speed.

The experimental protocol was described with details, but no figure of the experimental setup was included. Results need more clarification. The manuscript was written with good quality but some notations should be unified between the text, and figures. In the following, you can find few points that should be addressed:

1/ Line 112:  “Subjects were asked to find their preferred walking speed on a treadmill…”: please include the distance of the walking path in this paragraph.

To determine the preferred walking speed, we used the established protocol which was used in our previous studies. We added the below into the text.

“To determine the preferred walking speed, participants started walking at 0.67 m/s (1.5 mph). Then we asked them to let us know if we need to increase or decrease the speed based on their level of comfort. Once they indicated that they have reached a speed they are comfortable with, the speed was increased by 0.22 m/s (0.5 mph). They were asked to walk at this new speed for one minute and we asked their comfort level. If they indicated that the speed was too fast, the speed was slightly decreased (0.1 m/s). If they indicated that the faster speed was still comfortable, we increased the speed again (0.1 m/s). This process was repeated until they reach to a comfortable walking speed. To ensure that this is the correct speed, we asked them to walk for one minute, uninterrupted. If they approved that this is the preferred speed, we chose the speed. This process could take up to ten minutes. After the preferred speed was determined, the participants rested for five minutes.”

2/ Line 95-102: please add a figure including photos of the experimental setup. Figure 1 was added to the manuscript.

   Figure 1. An example of the experimental set up with one individual using the CAREN system.

3/ Line 134: “Only perturbations to the right side at left heel contact were used for data analysis”: Why the analysis was limited only to this side?

Thanks for the great comment. We aimed to determine the effect of medial perturbations which is defined as translating the walking platform to the right side at left heel contacts or translating the platform to the left side at right heel contacts (Contralateral). The responses to these two medial perturbations are similar according to the literature (1, 2). 

4/ Line 163-169: Please add figures to show examples of kinematics signals of the trunk, before, after and during the perturbation.

Thanks for the comment. We added the below figure (3), which is the linear and angular trunk velocity responses to the medium perturbation during walking at slow speed. The dashed line indicates the perturbation time.

Figure 3. Exemplar linear (left column) and angular (right column) trunk velocity trajectories in response to the medium perturbation during walking at slow speed. The dashed line indicates the perturbation time. Translation directions and axis of rotations were indicated on top of the linear and angular velocities, respectively.

5/ Line 165: “Therefore, velocity in six dimensions were calculated (d)”: what is the significance of “(d)” and is it used elsewhere in the manuscript?

We used “d” in equation 1. For each dimension we calculated the distance from steady state and then we calculate the sum of the distances under sigma.

6/ Line 197-184: please define each parameter in a separate line.

We modified the text to define each parameter in a separate line:

“D refers to the distance between the perturbed gait cycle and the average unperturbed limit cycle,

A refers to the relaxation distance,

B refers to the maximum distance after perturbation,

  refers to the time from the onset of perturbation to the maximum distance, and

β refers to the rate of return to the unperturbed cycle.”

7/ In Figure 4, Figure 5, table 3, table 4, table 5 and table 6: please unify and use the same symbols for the parameters tau (τ) and beta (β) as are used in the text.

We modified the tables and figures to address this comment.

References

  1. Roeles S, Rowe PJ, Bruijn SM, Childs CR, Tarfali GD, Steenbrink F, et al. Gait stability in response to platform, belt, and sensory perturbations in young and older adults. Medical & biological engineering & computing. 2018;56(12):2325-35.
  2. Hof AL, Vermerris SM, Gjaltema WA. Balance responses to lateral perturbations in human treadmill walking. The Journal of Experimental Biology. 2010;213:2655-64.

Round 2

Reviewer 2 Report

Thanks

Author Response

Thanks for your constructive feedback. We fixed the minor language/grammatical errors in the highlighted text of the manuscript.
